# Child's disability status and postnatal healthcare utilization among forcibly displaced women in Pakistan: A Secondary data analysis

Kalyani Dhar[1], Meshack Achore[1], Robert Kokou Dowou[2]*

1 Department of Population Health, School of Health Sciences, Hofstra University, Hempstead, New York, United States of America, 2 Department of Epidemiology and Biostatistics, School of Public Health, University of Health and Allied Science, Ho, Volta Region, Ghana

* 2017rdowou@uhas.edu.gh

## Abstract

The World Health Organization (WHO) estimates that about half of maternal and 40% of neonatal deaths occur within 24 hours of childbirth. Many women and newborns, especially among forcibly displaced populations, lack timely access to essential care. While research has examined antenatal and postnatal care (PNC) utilization in low-resource settings, little is known about displaced women raising children with disabilities. This study explores factors influencing PNC utilization among forcibly displaced women in Pakistan, focusing on displacement, maternal health, and childhood impairment. Using secondary data from the 2022 UNHCR Health and Utilization Survey (HAUS), this study examined the relationship between a child's disability and maternal PNC utilization among 2,847 displaced mothers aged 18–49 who had given birth within the past year. Multivariable logistic regression assessed how child impairment influenced PNC use while controlling for socioeconomic, demographic, and healthcare access factors. Among respondents, 65% utilized PNC services, and 17.6% reported having a child with an impairment. Most were married (85.9%), had completed primary education (88.9%), and delivered in public health facilities. Two-thirds had at least one antenatal care (ANC) visit. Child impairment significantly influenced maternal PNC utilization (OR = 1.24; 95% CI: 1.02–1.76). Mothers with tertiary education and children with impairments were less likely to use PNC compared with those with primary education (OR = 0.21; 95% CI: 0.09–0.52). ANC strongly predicted PNC use (OR = 9.99; 95% CI: 3.32–30.05). Other significant predictors included place of delivery, healthcare authorization, and ability to pay. Women who accessed antenatal care were nearly ten times more likely to use postnatal services. Findings highlight the need to strengthen ANC–PNC continuity and address financial, structural, and legal barriers limiting equitable, disability- and displacement-sensitive maternal healthcare in Pakistan.

**Data availability statement:** The data were stored in the publicly accessible repository of the UNCHR/World Bank (https://microdata.unhcr.org/index.php/catalog/810/related-materials). To access the data, individuals must complete a registration process designed explicitly for legitimate research purposes. The data repository provides full documentation, including sampling methodology and weighting procedures.

**Funding:** The authors received no specific funding for this work.

**Competing interests:** The authors have declared that no competing interests exist.

## Introduction

The postnatal period is the time immediately following a baby's birth, lasting up to six weeks. It is a particularly critical phase, as the health of both mother and newborn can be highly vulnerable. The World Health Organization (WHO) has noted that about 50% of maternal deaths and 40% of neonatal deaths happen within the first 24 hours after childbirth [1]. Despite this heightened risk, many women and newborns, especially in low-resource settings, lack timely access to healthcare during this period. Studies further show that three-quarters of neonatal deaths occur within the first week of the postnatal period [2], underscoring the life-saving importance of adequate postnatal care (PNC).

Globally, progress in reducing maternal mortality has been notable: between 2000 and 2023, the global maternal mortality ratio decreased by roughly 40%, falling from 328 to 197 deaths per 100,000 live births [1]. Despite these gains, in 2023 an estimated 260,000 women still died due to largely preventable complications related to pregnancy and childbirth, with a dramatic disparity between low- and high-income countries (346 vs. 10 per 100,000 live births) [1]. These figures highlight the fragility of maternal health systems, especially under threats like shrinking international aid, conflict, or pandemics.

Over the past two decades, global maternal and child health initiatives have prioritized antenatal care and skilled delivery attendance; however, PNC has often received less attention. Yet, the continuum of maternal and newborn care is incomplete without ensuring access to high-quality PNC. Evidence demonstrates that consistent PNC visits provide critical opportunities to detect maternal complications such as postpartum hemorrhage, infections, and pre-eclampsia, as well as neonatal conditions including sepsis, respiratory distress, and feeding difficulties. PNC also facilitates health education, family planning counseling, and essential newborn immunization, all of which contribute to reducing preventable morbidity and mortality.

Several studies have evaluated the utilization of PNC services across diverse settings. For example, a community-based cross-sectional survey of 798 married women in rural Western Ethiopia examined factors influencing PNC uptake. Findings indicated that maternal education, antenatal care attendance, and awareness of service benefits were associated with increased utilization. The study concluded that educating women and raising awareness about maternal health services during pregnancy substantially improves PNC use [2]. Similarly, a cross-sectional study of 600 mothers in Malawi identified maternal education, household income, decision-making power, knowledge of available services, awareness of postpartum danger signs, and place of delivery as significant predictors of PNC use [3]. In Nepal, Shrestha et al. [4] found that poor health communication was the primary barrier, with mothers lacking awareness about the timing and benefits of PNC. Collectively, these studies emphasize that education, awareness, and effective communication are crucial drivers of maternal health service utilization.

The primary goals of PNC services are to promote and maintain the well-being of both the woman and the newborn, while enabling healthcare providers to detect and manage potential complications promptly [5]. Despite its proven importance, PNC

PLOS **Global Public Health**

utilization remains limited in many low- and middle-income countries (LMICs), particularly in sub-Saharan Africa and South Asia. The literature has therefore focused heavily on identifying determinants of PNC use among general populations in LMICs [6,7].

However, far less is known about PNC utilization in humanitarian settings, despite the disproportionate health risks faced by forcibly displaced populations. Refugees and internally displaced persons often encounter unique and complex barriers to healthcare, including language differences, cultural unfamiliarity, lack of documentation, legal restrictions, and financial insecurity. Moreover, forcibly displaced families frequently require authorization from host governments to access care, which adds another layer of complexity.

To our knowledge, no study has specifically examined PNC uptake among forcibly displaced women raising children with disabilities (hereafter referred to as impairments). This represents a critical gap in global health research. Children with disabilities may require additional medical attention, and their mothers may face stigma, discrimination, and heightened stress, all of which can influence healthcare-seeking behaviors. It is therefore vital to understand whether child impairment status influences maternal utilization of PNC services in refugee populations, and to identify the enabling and inhibiting factors that shape these behaviors.

The present study addresses this gap by investigating PNC utilization among forcibly displaced women in Pakistan who are raising children with impairments. This focus is significant for two reasons. First, the factors shaping healthcare utilization in the general population are unlikely to fully explain the experiences of women in humanitarian contexts. Second, by examining the intersection of displacement, disability, and maternal health, this study generates evidence that can inform targeted interventions. Thus, this study aimed to explore the association between a child's disability status and postnatal healthcare utilization among forcibly displaced women in Pakistan, and to identify the socioeconomic and health-system factors that influence postnatal care use within this population. We conceptualize impairment as an exposure variable influencing postnatal care (PNC) utilization. This framing aligns with the social model of disability and the World Health Organization's International Classification of Functioning, Disability and Health (ICF), which views impairment as a structural determinant that shapes access to health services [8]. Women with children with physical, sensory, or cognitive impairments often face systemic, logistical, and attitudinal barriers that can limit their ability to seek and receive timely PNC. By examining impairment as the exposure, we aim to quantify its influence on PNC uptake and identify the health system and social barriers that contribute to inequities in postnatal service use among women with children with disabilities. Findings will provide valuable insights for policymakers, humanitarian organizations such as the World Bank and UNHCR, and host governments seeking to improve maternal and child health outcomes. Ultimately, addressing barriers to PNC use among displaced women with impaired children can contribute to reducing inequities in healthcare access and improving survival for both mothers and newborns.

## Methodology

### Study setting and data source

Pakistan hosts one of the world's longest-standing refugee populations, with over 1.75 million Afghan refugees officially registered within its borders (United Nations High Commissioner for Refugees [9]. Although current national policies allow refugees to access the public health system on the same terms as Pakistani citizens, disparities persist in the quality and continuity of maternal and newborn care. Evidence suggests that refugee women face significant barriers to accessing antenatal and postnatal care (PNC) services, including financial constraints, limited mobility, and inadequate awareness of available health resources. These barriers are particularly concerning given the high rates of neonatal morbidity and mortality within displaced communities, where complications during the immediate postpartum period often go undetected or untreated. The absence of standardized data on refugee maternal and child health outcomes has further hindered efforts to evaluate PNC service coverage, identify service gaps, and develop targeted interventions to improve neonatal survival and maternal well-being among forcibly displaced populations in Pakistan.

To address this gap, the United Nations High Commissioner for Refugees (UNHCR) implemented the Health and Utilization Survey (HAUS), a nationally representative survey conducted in 2022 across major Afghan refugee-hosting areas in Pakistan. Implemented in collaboration with the Government of Pakistan's Commissionerate for Afghan Refugees (CAR) and the World Bank, the survey covered refugee settlements and urban host communities in Khyber Pakhtunkhwa, Balochistan, and Punjab, the three provinces hosting over 90% of registered Afghan refugees. HAUS employed a stratified sampling approach to ensure representativeness by province, settlement type, and household composition. Data collection included demographics, healthcare access and utilization, barriers, and maternal and child health outcomes. The dataset comprises over 8,000 refugee households, making it one of the most comprehensive sources of information on refugee health in Pakistan. The survey collected detailed information on respondents' understanding of healthcare services, actual utilization, barriers to access, and perceived quality of care. Importantly, HAUS provides one of the most reliable and nationally representative datasets available on refugee health in Pakistan, making it uniquely suited for studying maternal and child health dynamics in this population. Data for this study were accessed on June 12, 2024. The data were stored in the publicly accessible repository of the UNCHR/World Bank (https://microdata.unhcr.org/index.php/catalog/810/related-materials). To access the data, individuals must complete a registration process designed explicitly for legitimate research purposes. The data repository provides full documentation, including sampling methodology and weighting procedures.

## Maternal health among Afghan refugees in Pakistan

Historic data show that Afghan refugee women in Pakistan have faced dramatically elevated risks of maternal death. In settlements in the Hangu district, the maternal mortality ratio (MMR) was estimated at 291 per 100,000 live births, substantially higher than national averages, yielding a lifetime maternal death risk of 1 in 50 (8). Interventions by the International Rescue Committee (IRC) that expanded emergency obstetric care and improved community health knowledge contributed to reducing this MMR to 102 per 100,000 by 2004 [10]. These figures contextualize the structural vulnerabilities faced by refugee mothers, even before accounting for the added complexities of caring for children with disabilities and underscore the urgent need for targeted policies.

## Ethics statement

This is a secondary data analysis of a publicly available database. This data is de-linked, anonymised, and cannot be traced back to individual participants. Thus, there was no violation of the participants' confidentiality. For the present secondary analysis, additional ethics approval was not required because only anonymized data were used. Access to the HAUS dataset is granted through the World Bank/UNHCR microdata repository following a formal registration and approval process. Researchers are required to comply with UNHCR's data protection standards before receiving access.

## Study design, population, and sample size

This study employed a cross-sectional secondary data analysis using data from the 2022 UNHCR Health and Utilization Survey (HAUS). A subset (n = 2847) of the total sample size of 8,264 was used for our analysis, as we were interested in women aged 18–49 years who had given birth within the 12 months preceding the survey. This inclusion criterion ensures the study population aligns with the World Health Organization's definition of the postnatal period (up to six weeks after childbirth). Respondents without recent childbirth or with incomplete data on postnatal care use were excluded.

## Dependent variable

The dependent variable for this study was postnatal care (PNC) utilization. According to the World Health Organization [11], the postnatal period refers to the six weeks following childbirth, a critical phase for the health and survival of both mother and newborn. Postnatal care includes the assessment of maternal and neonatal health, detection and

management of complications, counseling on breastfeeding, family planning, and hygiene, as well as immunization services. In the HAUS dataset, PNC utilization was measured based on the question: *"Did you receive postnatal care services following your most recent delivery?"* Responses were coded dichotomously as Yes (1) for women who reported receiving PNC and No (0) for those who did not. This operational definition aligns with WHO's global maternal health monitoring framework and ensures comparability with other studies in low- and middle-income country contexts.

### Primary independent variable

The primary independent variable was child impairment status, defined in accordance with the World Health Organization's International Classification of Functioning, Disability and Health (ICF) framework [8]. In this study, *impairment* refers to any physical, sensory, or cognitive limitation in a child as reported by the mother at the time of the survey. Mothers were asked whether any of their children had a condition or limitation that affected movement, sight, hearing, speech, learning, or behavior. Because the HAUS survey did not restrict the timing of the impairment's onset, reported cases may reflect conditions recognized at birth or during early infancy. Conceptually, we treated impairment as the exposure variable since it represents a child-level factor that can shape maternal healthcare-seeking behavior, particularly postnatal care utilization. This framing aligns with the social model of disability, which recognizes impairment as a determinant of access and interaction within health systems, rather than as a health outcome. Treating impairment as an exposure, thus, enables a data-driven assessment of how disability status translates into differential access to essential maternal health services. This perspective advances the literature by moving beyond descriptive accounts of vulnerability to provide empirical evidence on the extent and determinants of inequities in PNC utilization among women with children with disabilities in resource-limited settings.

### Other independent variables

Other independent variables were selected based on prior evidence in the maternal health literature and included demographic, socioeconomic, and healthcare-related characteristics. These comprised maternal age, gender of the child, employment status, marital status, immunization history (schedule adherence and card possession), antenatal care attendance, number of ANC visits, type of delivery, and financial factors such as whether the family paid for delivery. The categorization of these variables is presented in Table 2, which includes a broad set of covariates that allow us to control for potential confounders and isolate the association between child impairment status and PNC utilization.

### Statistical analysis

This study used data from the United Nations High Commissioner for Refugees (UNHCR) Health and Access Utilization Survey (HAUS). The Health and Access Utilization Survey (HAUS) dataset includes data from both male and female respondents across refugee households. For this analysis, we restricted the sample to women of reproductive age (18–49 years) who had given birth within the 12 months preceding the survey, aligning with the World Health Organization's definition of the postnatal period. Respondents without recent childbirth or with missing data on key variables were excluded. The final analytic sample consisted of 2,847 forcibly displaced women. Male respondents and women without recent childbirth were excluded from the analysis to ensure conceptual and analytical consistency with the study's objective of examining postnatal healthcare use among mothers raising children with impairments.

Impairment was treated as the primary exposure variable, consistent with the WHO International Classification of Functioning, Disability and Health (ICF) framework, recognizing that physical, sensory, or cognitive impairments can affect women's ability to access and utilize postnatal care (PNC) due to mobility limitations, stigma, or systemic barriers. Modeling impairment as the exposure variable allowed for assessing how disability status influences PNC uptake and identifies inequities in service utilization.

All variables were assessed for missingness and internal consistency. Missing data were minimal (<5%) and handled through listwise deletion (complete case analysis). Given the low missingness, no imputation procedures were applied.

All analyses were conducted using **Stata version 18 (StataCorp LLC, College Station, TX, USA)**. The analysis proceeded in three steps:

1. Descriptive statistics were used to summarize sample characteristics using frequencies and percentages for categorical variables and weighted means with standard deviations for continuous variables.

2. Univariable logistic regression was used to examine the bivariate association between each independent variable and PNC utilization. Variables with $p \leq 0.20$ were retained for multivariable analysis.

3. Multivariable logistic regression was used to assess the adjusted association between child impairment and PNC utilization, controlling for sociodemographic and healthcare-related covariates.

   ○ *Model 1:* Unadjusted association between child impairment and PNC use.

   ○ *Model 2:* Adjusted for confounders.

Model performance was evaluated using the Akaike Information Criterion (AIC), Bayesian Information Criterion (BIC), and Hosmer–Lemeshow chi-square test ($p > 0.05$), indicating good fit. Multicollinearity was tested using Spearman's correlation (threshold > 0.70), and outliers were examined using chi-square distribution diagnostics. No violations were detected. Model explanatory power was assessed via Cox–Snell $R^2$ and Nagelkerke $R^2$. All associations were reported as odds ratios (ORs) with 95% confidence intervals (CIs), and significance was set at $p < 0.05$, with $0.05 \leq p \leq 0.10$ considered trends.

## Results

### Sample characteristics

Table 1 presents the descriptive statistics of the study participants. Sociodemographic and clinical characteristics of study participants. A total of 2847 participants were included in the analysis. Most participants were aged 18–29 years (51.2%), followed by those aged 30–39 years (33.6%), and those aged 40 years or older (15.3%). Regarding marital status, a substantial proportion (85.9%) were married, while 14.1% were not married. In terms of education, the majority of respondents (88.9%) had primary-level education, while 7.2% attained tertiary education, and smaller proportions completed middle (2.9%) or secondary (1.0%) education. For maternal healthcare utilization, 64.9% reported attending antenatal care (ANC) during pregnancy, while 35.0% did not. Over half (54.6%) had fewer than four ANC visits, and 45.5% had more than four.

### Association between postnatal care service utilization among parents with children with impairment and independent variables/ regression

Table 2 shows the results of bivariate, univariate, and multivariable logistic regression analyses examining the link between postnatal care use among parents with children who have impairments and various predictor variables. Model 1 explores the relationship between child impairment status and postnatal care utilization. Our findings suggest that parents whose children were reported to have some form of impairment after birth are more likely to use postnatal services (OR= 1.36, CI = 1.03-1.80) compared to those without impairments.

Model 2 presents a multivariable logistic regression analyzing the association between postnatal care use among parents with children with impairments. Even after adjusting for other factors, the results show a significant link between child impairment and maternal postnatal care use (OR= 1.24, CI = 1.02-1.76). Parents with children with impairments who had tertiary education (OR= 0.21, CI = 0.09-0.52) were less likely to use postnatal care than those with primary education.

**Table 1. Descriptive statistics.**

| Variables | Frequency (%) |
|---|---|
| **Gender** | |
| Females | 3,751(45.39%) |
| Males | 4,513(54.61%) |
| **Age** | |
| 18-29years | 1,456(51.15%) |
| 30-39years | 955(33.55%) |
| 40+years | 436(15.3%) |
| **Marital Status** | |
| Married | 2,445(85.89%) |
| Not married | 402(14.10%) |
| **Immunization card** | |
| No | 55(1.93%) |
| Yes | 2792(98.06%) |
| **Measles vaccine** | |
| No | 63(2.22%) |
| Yes | 2784(97.77%) |
| **Polio vaccine** | |
| No | 14(0.50%) |
| Yes | 2833(99.50%) |
| **Educational level** | |
| Primary | 2532(88.94%) |
| Middle | 83(2.90%) |
| Secondary | 28(0.98%) |
| Tertiary | 204(7.17%) |
| **Antenatal care use** | |
| No | 997(35.01%) |
| Yes | 1850(64.99%) |
| **Number of antenatal care visits** | |
| 0-3 times | 1,553(54.55%) |
| >3 times | 1294(45.46%) |
| **Type of delivery** | |
| Caesarean planned | 175(6.12%) |
| Caesarean urgent | 131(4.60%) |
| Vaginal normal | 2541(89.28%) |
| **Paid for delivery** | |
| No | 2193(77.02%) |
| Yes | 654(22.98%) |
| **Impaired child** | |
| No | 2345(82.37%) |
| Yes | 502(17.63%) |
| **Required healthcare authorization** | |
| No | 2441(85.74%) |
| Yes | 406(14.26%) |

**Table 2.** Bivariate and multivariable analysis of postnatal service utilization among parents with children with impairment.

| Independent Variables | Model 1<br>Postnatal care utilization. OR(CI) | Model 2<br>Postnatal care utilization. OR(CI) |
|---|---|---|
| **Impairment** | | |
| No | Ref | |
| Yes | 1.36(1.03-1.80)** | 1.24(1.02-1.76)** |
| **Educational level** | | |
| Primary | | Ref |
| Middle | | 0.43(0.16-1.15) |
| Secondary | | 0.84(0.20-3.44) |
| Tertiary | | 0.21(0.09-0.52)** |
| **Marital status** | | |
| Single | | Ref |
| Married | | 0.05(0.00-4.11) |
| **Age** | | |
| 18-29 | | Ref |
| 30-39 | | 1.5(0.09-26.01) |
| 40-49 | | 1.25(0.05-26.86) |
| **Antenatal care** | | |
| No | | Ref |
| Yes | | 9.99(3.32-30.05)** |
| **Frequency of antenatal care** | | |
| 1-2 times | | Ref |
| More than 3 times | | .83(0.65-1.02) |
| **Face difficulties accessing antenatal care** | | |
| No | | Ref |
| Yes | | .26(.09-0.79)* |
| **Paid for delivery** | | |
| No | | Ref |
| Yes | | 1.11(0.74-1.65) |
| **Place of delivery** | | |
| Public hospital | | Ref |
| Home_No_skilled_attendant | | 0.75(0.40-0.98)** |
| Home_With_skilled_attendant | | 0.49(0.13-1.10) |
| Private_hospital | | 1.28(1.02-2.94)* |
| **Healthcare authorization** | | |
| No | | Ref |
| Yes | | 2.85(1.01-8.09)* |
| **Accessed 1st healthcare** | | |
| No | | Ref |
| Yes | | 3.02(1.21-11.12)* |
| **Polio vaccine** | | |
| No | | Ref |
| Yes | | 1.13(0.23-5.67) |
| **Immunization schedule** | | |
| No | | Ref |

*(Continued)*

**Table 2.** (Continued)

| Independent Variables | Model 1<br>Postnatal care utilization. OR(CI) | Model 2<br>Postnatal care utilization. OR(CI) |
|---|---|---|
| Yes | | 1.56(.36-6.71) |
| **Frequency of antenatal use** | | |
| Less than 3 times | | Ref |
| More than 3 times | | .82(.65- 1.05) |
| **Pay for delivery** | | |
| No | | Ref |
| Yes | | 1.56(1.36-6.71)** |
| **Cons** | .019(0.01- 0.035) | .01(0.09-69.55) |
| **AIC** | 789 | 620 |
| **BIC** | 804 | 889 |

Odds Ratios (OR) and 95% Confidence Intervals (CI) are reported. Asterisks denote statistical significance: *p < 0.05, **p < 0.01, ***p < 0.001. Model fit statistics: Hosmer–Lemeshow $\chi^2$ = 7.82 (p = 0.45), Cox–Snell $R^2$ = 0.18, Nagelkerke $R^2$ = 0.27.

Parents who utilized antenatal care were more likely to then use postnatal care (OR = 9.99, CI = 3.32-30.05). Those who faced challenges accessing antenatal care (OR = 0.26, CI = 0.09-0.79) were less likely to use postnatal care. The place of delivery also appeared necessary: individuals who delivered at home without a skilled attendant were less likely to use postnatal care (OR= 0.75, CI = 0.40-0.98), while those who delivered at a private hospital were more likely to access postnatal care (OR= 1.28, CI = 1.02-2.94) compared to deliveries at a public hospital. If a person had healthcare authorization, they were more likely to utilize postnatal care (OR = 2.85, CI = 1.01-8.09). Parents who paid for delivery (OR= 1.56, CI = 1.36-6.71) were also more likely to access postnatal services than those who did not.

The final multivariable logistic regression model demonstrated a satisfactory fit, with a Hosmer–Lemeshow chi-square of 7.82 and a p-value of 0.45, indicating no evidence of poor fit. The model explained a meaningful portion of the variance in postnatal care utilization (Cox–Snell $R^2$ = 0.18; Nagelkerke $R^2$ = 0.27). AIC and BIC values were 620 and 889, respectively, further confirming the parsimony of the final model.

Participants indicated their inability to pay for care (38%) (Fig 1) is the most cited reason for not using postnatal care services. This is followed by the distance, as people (36%) who perceive the distance to be too far are more likely to refuse or underutilize the postnatal services. Long waiting time was another reason participants would not take up post-natal healthcare services. The remaining factors also play a critical role in maternal postnatal services.

## Discussion

The current study examines factors that affect the use of postnatal care services among forcibly displaced women in Pakistan who have children with disabilities. Since postnatal care can significantly reduce preventable maternal and child mortality, its utilization is crucial. Although a few studies have looked into postnatal care use among mothers with disabilities [12–14], no research has investigated how a newborn's impairment status influences postnatal care use, especially among women in humanitarian settings. The importance of this study is twofold. First, the factors that influence the general population's use of postnatal care services are likely different from those affecting women in humanitarian contexts. Second, women with children with impairments are often assumed to be less likely to use postnatal care due to stigma. Therefore, it is essential to first test this assumption and explore factors that either encourage or hinder postnatal care use among this specific group. Our analytical framing of impairment as the exposure variable is intentional and rooted in

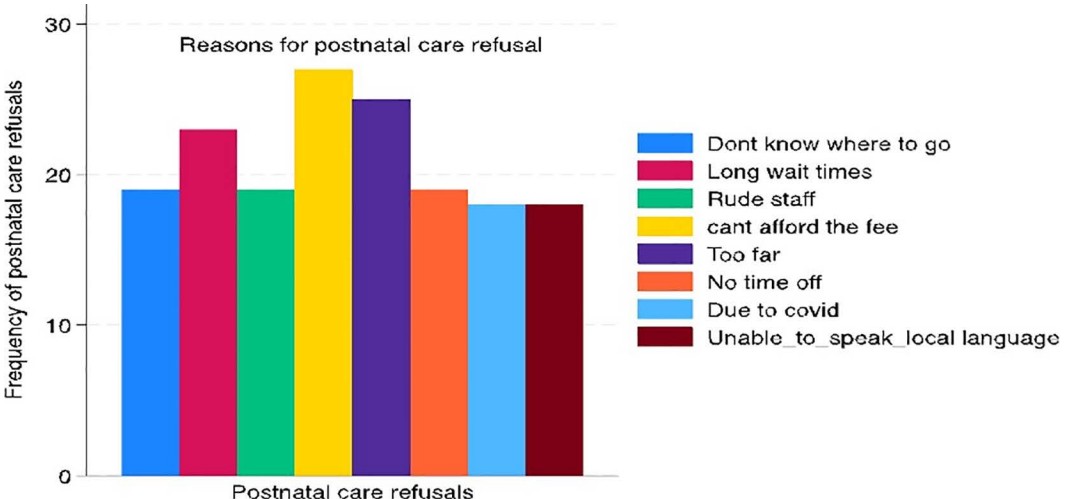

**Fig 1. Reasons for postnatal care refusals.**

the broader determinants of health equity framework. Impairment often precedes and conditions women's engagement with healthcare systems, influencing mobility, autonomy, and provider interaction, all of which affect postnatal care utilization. Treating impairment as an exposure, therefore, enables a data-driven assessment of how disability status translates into differential access to essential maternal health services. This perspective advances the literature by moving beyond descriptive accounts of vulnerability to provide empirical evidence on the extent and determinants of inequities in PNC utilization among women with children with disabilities in resource-limited settings.

Our results show that mothers whose children had some form of impairment after birth were more likely to utilize postnatal care (PNC) services, which contrasts with the common assumption that disability is associated with lower healthcare access. Mothers of children with impairments often recognize that their child requires more medical attention, monitoring, or specialized care. This awareness motivates them to seek postnatal checkups and follow-up services more diligently, ensuring the child's condition is evaluated, managed, or referred appropriately. Beyond that, several explanations may account for this finding. First, infants with visible or suspected impairments are often flagged for follow-up care by healthcare providers, prompting mothers to return for postnatal or pediatric visits. Second, it is possible that some impairments were identified during postnatal consultations, resulting in a positive association between impairment and PNC use due to reverse causality rather than a direct causal effect. Third, mothers of infants with early health complications or developmental concerns may receive mandatory or advised health checks from providers, increasing their likelihood of PNC attendance. In addition, although most forcibly displaced people require authorization from host governments to access healthcare, our findings suggest that parents often prioritize their children's health needs by engaging in postnatal services when possible. Research indicates that, despite unmet needs, children with disabilities are generally more likely to seek healthcare compared to their peers without disabilities [14]. This suggests a broader pattern of increased health system engagement among families facing health vulnerabilities. Moreover, postnatal care services, including genetic counseling, immunization, and growth monitoring, play a critical role in the primary prevention of childhood disabilities [15]. Taken together, these mechanisms suggest that the observed association may partly reflect the health system's responsiveness to neonatal complications rather than a purely behavioral response by mothers. Nonetheless, the findings underscore that PNC encounters are essential opportunities for early support for children with impairments in humanitarian contexts. Future longitudinal and qualitative studies are needed to clarify the temporal and causal relationship between child impairment, diagnosis, and healthcare utilization among forcibly displaced women.

 

The current study finds that women with children who have impairments and who use antenatal care services are also more likely to utilize postnatal care services. This finding supports other studies [16–18]. For example, a study by 17. Atuhaire et al. [17] in Uganda demonstrated that early use of antenatal care led to deliveries at health facilities (such as hospitals, clinics, and community health centers) and increased use of postnatal care. Chaka et al. [18] investigated how often and what factors influence postnatal care utilization in Ethiopia. They found that mothers who had previously used antenatal care and had more than two visits were more likely to seek postnatal care. Similarly, a study by Abota et al. [19] in Ethiopia showed that married women who attended antenatal care (ANC) were more likely to participate in postnatal care than those who did not attend ANC.

Reasons for refusing postnatal care include poor communication and lack of information. For example, a study in Kathmandu, Nepal, examining barriers to mothers' use of postnatal services, found that poor communication was a major obstacle. When antenatal care is used during pregnancy, issues with communication and information are reduced, as the importance of postnatal care is stressed during these visits. It's also important to note that in humanitarian settings where permission is needed to access care, the link between antenatal and postnatal care might not exist, even if the willingness to seek care is present. This is mainly because access depends on the host countries. Host countries might grant access during pregnancy but deny it after birth. Therefore, we recommend that women in these settings be given long passes to use healthcare services during and after pregnancy. Additionally, those who pay for delivery are more likely to use postnatal care. The possible reason is that people who can afford private health services in humanitarian contexts are either financially well-off or educated enough to understand the importance of delivering at a health facility. These same factors often motivate them to seek postnatal care, especially if their newborn has an impairment.

Similarly, we found that women who delivered in a private hospital, where they paid for services, were more likely to use postnatal care than those who delivered in a public hospital or at home. This aligns with previous research examining the relationship between place of delivery and postnatal care utilization. For example, a study by Ayele et al. [20] showed that the use of postnatal care services among mothers who delivered at home is low. The study found that mothers living in urban areas, following the Orthodox religion, with a higher wealth index, possessing a bank account, using the calendar method to delay pregnancy, perceiving the pregnancy as wanted, attending four or more antenatal care visits, and listening to the radio at least weekly were all linked to higher postnatal care utilization. This underscores the importance of targeted strategies to improve socio-economic status, enhance the continuum of care, and boost health literacy to increase postnatal care use among women delivering at home. Another study in Nigeria by Dahiru et al. [21] indicated that antenatal care is a strong predictor of both health facility delivery and postnatal care utilization, while health facility delivery also predicts postnatal care. In Zambia, it was concluded that women who delivered in a health facility were more likely to utilize postnatal care [22] compared to those who delivered at home. Although not explicitly focused on children with disabilities, the study emphasized that the place of delivery plays a crucial role in postnatal care utilization.

Although several studies [21,23] argue that educational achievement is connected to postnatal care and healthcare system utilization. Interestingly, our results indicate that women with tertiary education were less likely to utilize PNC services compared to those with only primary education. Although this appears counterintuitive, the result may reflect the complex interplay between structural barriers and displacement-related constraints rather than education itself. Forcibly displaced women with higher education often face employment restrictions, limited autonomy, or administrative barriers in host-country health systems that constrain healthcare access. This finding aligns with research suggesting that education alone does not translate into health service use when socioeconomic, legal, or geographic obstacles persist in humanitarian settings.

Healthcare access plays a significant role in postnatal care utilization. We found that forcibly displaced individuals with healthcare authorization from the host country were more likely to use postnatal care services. If a patient has prior healthcare authorization, they are more likely to utilize the services because they understand what is available and can be used. A study by Vidler et al. [24] in rural South Indian communities reported regular use of healthcare services during

pregnancy and for delivery. The uptake of maternity care services was attributed to new government programs and increased availability of maternity services, which shows that mothers with access to healthcare are most likely to utilize postnatal care. Another study in northwest Ethiopia by Zeleke et al. [25] showed low postnatal care utilization. The study discussed the importance of personal health education, relevant infrastructure, and transportation to reduce barriers to postnatal care use. Similarly, first-time healthcare users were more likely to utilize postnatal care services. If this is their first time using healthcare, they probably just received authorization from the host country to access care, which motivates and encourages them to use their newly granted access. Overall, those with healthcare access are more likely to utilize postnatal care.

Our findings must be interpreted within the broader realities of humanitarian displacement, where legal status and healthcare authorization significantly influence access to maternal healthcare. In Pakistan, Afghan refugees and other forcibly displaced women are required to obtain formal authorization to access national health facilities, which can delay or discourage healthcare-seeking, particularly during the postnatal period. This barrier is compounded by mobility restrictions, dependency on host-country health systems, and the absence of consistent health insurance coverage. Women who secure authorization or pay for private delivery services are more likely to access PNC, suggesting that legal and financial gatekeeping mechanisms play a decisive role in determining care continuity after childbirth.

Unlike women in stable low-resource communities, displaced mothers navigate complex institutional pathways where healthcare access is conditional on administrative approval or donor-funded programs. This structural dependency distinguishes humanitarian populations from other LMIC contexts and may explain the observed disparities in PNC utilization. Our results therefore contribute to the growing literature emphasizing that humanitarian health inequities are produced not only by poverty or education, but also by governance structures and policy regimes that regulate refugee health rights and entitlements. Addressing these barriers requires policy coordination between host governments, UNHCR, and development partners to ensure that maternal and postnatal services are accessible regardless of legal status.

## Limitations

The study has notable limitations. Besides the inherent issues with secondary data, it relied on questions about past activities, which increases the risk of recall bias. There is also a possibility of selection bias since those negatively affected by a lack of healthcare access might be more inclined to participate than others. Additionally, since the data is self-reported, it may be subject to biases such as overreporting or underreporting. Our results show that mothers of children reported to have impairments were more likely to utilize postnatal care (PNC) services. However, because impairments were identified based on maternal reports at the time of the survey, it is not possible to determine whether these impairments were present at birth or recognized later. It is therefore plausible that the association reflects reverse temporality, where PNC use facilitates the identification or diagnosis of impairments rather than the impairment motivating PNC utilization.

## Conclusion

This study examined factors influencing postnatal care (PNC) utilization among forcibly displaced women in Pakistan, raising children with impairments. We found that women who attended antenatal care (ANC) wer significantly more likely to use PNC services, underscoring the importance of strengthening the continuum of maternal care from pregnancy through the postnatal period. Conversely, women with tertiary education were less likely to utilize PNC, suggesting that structural barriers, such as displacement-related restrictions, limited healthcare authorization, and financial dependence, can outweigh the advantages of education. The study also revealed that healthcare authorization and the ability to pay for delivery strongly influenced PNC uptake, highlighting inequities embedded in the administrative and financial systems governing refugee healthcare access. Policy efforts should prioritize integrated ANC–PNC programs, legal and financial inclusion for displaced women, and disability-sensitive maternal health services. Strengthening coordination among host

governments, UNHCR, and development partners will be essential to ensure that postnatal services are accessible, equitable, and inclusive for all displaced mothers and children.

## Author contributions

**Conceptualization:** Kalyani Dhar.

**Data curation:** Kalyani Dhar, Meshack Achore.

**Formal analysis:** Kalyani Dhar, Meshack Achore, Robert Kokou Dowou.

**Investigation:** Kalyani Dhar, Meshack Achore.

**Methodology:** Kalyani Dhar, Meshack Achore, Robert Kokou Dowou.

**Project administration:** Kalyani Dhar, Meshack Achore.

**Software:** Kalyani Dhar.

**Supervision:** Kalyani Dhar, Meshack Achore.

**Validation:** Meshack Achore, Robert Kokou Dowou.

**Visualization:** Meshack Achore, Robert Kokou Dowou.

**Writing – original draft:** Kalyani Dhar, Meshack Achore.

**Writing – review & editing:** Kalyani Dhar, Meshack Achore, Robert Kokou Dowou.

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
