## [Decision Letter · Decision Letter 0]

27 Oct 2025

PGPH-D-25-02580

Child's disability status and postnatal healthcare utilization among forcibly displaced women: Evidence from Pakistan

Dear Dr. Dowou,

Thank you for submitting your manuscript to PLOS Global Public Health. After careful consideration, we feel that it has merit but does not fully meet PLOS Global Public Health’s publication criteria as it currently stands. Therefore, we invite you to submit a revised version of the manuscript that addresses the points raised during the review process.

We look forward to receiving your revised manuscript.

Kind regards,

Ejemai Eboreime, MD, MSc, PhD

Academic Editor

Journal Requirements:

1. We have amended your Competing Interest statement to comply with journal style. We kindly ask that you double check the statement and let us know if anything is incorrect.

2. Please provide a/amend your detailed Financial Disclosure statement. This is published with the article. It must therefore be completed in full sentences and contain the exact wording you wish to be published.

3. We note that your Data Availability Statement is currently as follows: “All the data are available within the manuscript”

Additional Editor Comments (if provided):

Reviewers' comments:

Reviewer's Responses to Questions

**Comments to the Author**

1. Does this manuscript meet PLOS Global Public Health’s publication criteria?

Reviewer #1: Partly

Reviewer #2: Yes

2. Has the statistical analysis been performed appropriately and rigorously?

Reviewer #1: No

Reviewer #2: Yes

3. Have the authors made all data underlying the findings in their manuscript fully available (please refer to the Data Availability Statement at the start of the manuscript PDF file)?

Reviewer #1: Yes

Reviewer #2: Yes

4. Is the manuscript presented in an intelligible fashion and written in standard English?

Reviewer #1: Yes

Reviewer #2: Yes

Reviewer #1: Review of “Child's disability status and postnatal healthcare utilization among forcibly displaced women: Evidence from Pakistan”

Date: 18/10/2025

Overall Comment

This is a good effort overall, and I appreciate the attempt to use national data to explore an important topic.

However, the study’s framing limits its relevance and interpretive value. The current direction of the analysis, treating impairment as the exposure and postnatal care (PNC) uptake as the outcome, feels counterintuitive and does not add much beyond what could be speculated without data.

It may be more meaningful to reverse the relationship, using PNC as the exposure variable. This would allow the study to explore whether PNC contributes to early diagnosis of neonatal impairment or, among children beyond the neonatal period, whether PNC influences the likelihood of disability. Additionally, the analytic process and sample selection are unclear, particularly how the author moved from the overall sample of 8,264 respondents to the subset used for analysis.

Here are more detailed comments for each section:

Title

It should be specified in the title that this is a secondary data analysis. Also would be good to include the timeframe in the title

Abstract

Methods: Start from the study type – it is a secondary data analysis. Also missing the sample size, time frame, inclusion criteria (e.g. mothers who gave birth in the past year?). Please clarify what your outcome variable is. I cant tell if its disability status, PNC utilization or even both.

Result: Start with sample characteristics (briefly) - what percentage of women used postnatal care? What proportion had children with impairments?

“.....Postnatal care (OR= 0.9.99, CI= 3.32-30.05)” The OR here seems wrong... did you mean 9.99? Also, it would be good to add p values, although the CIs hint at the significance

The direction of association for tertiary education seems counterintuitive. The authors should interpret why higher education correlates with lower use of care or verify if it’s a misclassification.

Conclusion: The conclusion repeats general awareness messaging rather than specific interpretations and recommendations from their study. Recommendations should directly follow from the statistical findings (e.g. targeting the educational disparities you found or advocating for continuity of ANC-PNC use etc).

Background

1. Be consistent with the bracket types used for citations

2. Add a clear objective statement at the end of the introduction for easy accessibility

Methods

Setting

1. “Pakistan currently faces one of the longest-running refugee crises in the world, with over 1.3 million Afghan refugees officially registered within its borders” ... this sentence should be referenced

2. More info is needed to be able to assess the representativeness of the HAUS dataset. Either you include a reference to the database, where it is properly described, or you give your readers a sense of the geographic coverage of the survey.

3. The authors should include when the data was collected and the sample scope

Maternal health...

4. Since this study focuses on PNC, this section should also focus on neonatal health and PNC services among this population

Dependent Variable

5. First include the definition of PNC here, so your readers are sure that it is the same as the global standards.

6. Having just yes/no may does not specify when, where (facility/community), the provider type, which I think are important variables when assessing PNC utilization.

Primary independent variable

7. Please clarify what you mean by impairment, as most disabilities are not typically apparent during the neonatal period. From your description, it appears that you used data based on mothers’ reports of whether their child had a disability during the neonatal period, and then analyzed this in relation to postnatal care (PNC) uptake. However, conceptually, it seems more appropriate for PNC uptake to serve as the independent variable in this relationship.

Statistical analysis

8. You should always include the company and place when mentioning statistical software - Stata version 18 (StataCorp LLC, College Station, TX, USA)

9. Was there missing data? And if yes, how did you treat it?

Results

1. Authors should start by telling us the number of participants

2. The sample description and analytical pathway are quite unclear. It is difficult to understand how the sample was derived and how it aligns with the study objectives.

• The total sample size of 8,264 respondents appears to include both men and women, but the study’s focus (postnatal care utilization among parents of children with impairments) suggests that only a specific subset, like women of reproductive age or those who recently gave birth (within a specified period, should have been analyzed. You need to clearly define who was included in this analysis and why.

• Please specify the inclusion criteria, for example: “Women who gave birth within the last 12 months” or “Mothers of children aged under five.”

Then, state the sample size for that subset and use that as your analytical sample, not the entire dataset.

• It is not clear how the analysis moved from the initial 8,264 respondents to the smaller numbers used in regression. You need to explain any exclusions or missing data and show the flow of the sample (perhaps in a short paragraph or figure).

• Without clarity on the sample used, the validity of the results especially the odds ratios and discussion, cannot be properly assessed.

3. In summary, please:

• Clearly define your study population.

• Report the final sample size after applying inclusion/exclusion criteria.

• Ensure that only the relevant subgroup (e.g., mothers who recently gave birth) is analyzed.

• Provide a short explanation of how the data were cleaned or filtered before analysis.

• Include these in the methodology section

4. What is the sample size that you used for your model???

5. Your Model 1 result can easily be reported as a narrative, so as to reduce the complexity of the table

6. Please add actual p values to the tables, not just the asterisks. Also, the asterisks are not labelled

7. In addition to AIC and BIC, please include model fit statistics such as the Hosmer–Lemeshow goodness-of-fit test and pseudo R² values (e.g., Cox–Snell or Nagelkerke). These help assess how well the logistic regression model explains and fits the data.

8. It remains unclear how “child impairment” was defined or measured. This is crucial for interpreting results and understanding the validity of your results.

Discussion

1. The central finding in this study, that mothers of impaired children were more likely to use PNC, is not adequately explained. The discussion speculates on emotional motivation (overwhelmed mothers), which I think is a week argument. The authors should consider alternative explanations like health complications prompting more follow-up visits, mandatory health checks for infants with visible conditions, or even reverse causality if impairment was diagnosed during a postnatal visit.

2. The manuscript still reads as if child impairment precedes postnatal care utilization. I think this conceptually problematic, since most impairments are not detected immediately at birth. The authors need to clarify whether impairment was identified postnatally and, if so, reframe the interpretation to reflect that PNC may predict impairment identification, not the other way around.

3. The discussion cites studies from Uganda, Ethiopia, and Nepal but does not connect the findings to humanitarian contexts, which was the paper’s stated contribution to the field. It should highlight how displacement, legal status, and healthcare authorization shape maternal care or PNC utilization patterns.

Conclusion

1. This is way too long. It should highlight the key findings, the implications and the key recommendations, written succinctly.

Reviewer #2: Comments

1. Primary Independent Variable: Child impairment: Lacks operational definition, what threshold determines impairment?

2. Dependent Variable definition: "Since delivery" is vague, no timeframe specified (within 48 hours? 6 weeks?)

3. No description of what constitutes PNC services (checkup only? Multiple visits? Specific components?)

Recommendations:

1. Specify the impairment assessment criteria

2. Report types and distribution of impairments (physical, cognitive, sensory)

3. Define the specific timeframe for PNC (e.g., "within 42 days postpartum")

4. Describe what services were included in the PNC definition

**Do you want your identity to be public for this peer review?** For information about this choice, including consent withdrawal, please see our Privacy Policy

Reviewer #1: No

Reviewer #2: No

---

## [Decision Letter · Decision Letter 1]

11 Dec 2025

PGPH-D-25-02580R1

Child's disability status and postnatal healthcare utilization among forcibly displaced women: Evidence from Pakistan

Dear Dr. Dowou,

Thank you for submitting your manuscript to PLOS Global Public Health. After careful consideration, we feel that it has merit but does not fully meet PLOS Global Public Health’s publication criteria as it currently stands. Therefore, we invite you to submit a revised version of the manuscript that addresses the points raised during the review process.

Please could you address a minor structural issue with this submission? We have noted that you have provided the track marked changed version, however, the manuscript file appears to be the original submission, and not the revised version. Could you please revise your submission so that the most recent version of your study is included as the manuscript file, without any trackmarked changes? Thank you very much for your attention to this request.

We look forward to receiving your revised manuscript.

Kind regards,

Johanna Pruller, Ph.D.

PLOS Staff Editor

Journal Requirements:

Reviewers' comments:

Reviewer's Responses to Questions

**Comments to the Author**

Reviewer #2: All comments have been addressed

publication criteria?

Reviewer #2: Yes

3. Has the statistical analysis been performed appropriately and rigorously?

Reviewer #2: Yes

4. Have the authors made all data underlying the findings in their manuscript fully available (please refer to the Data Availability Statement at the start of the manuscript PDF file)?

Reviewer #2: Yes

5. Is the manuscript presented in an intelligible fashion and written in standard English?

Reviewer #2: Yes

Reviewer #2: All my previous comments have been duly addressed

**Do you want your identity to be public for this peer review?** For information about this choice, including consent withdrawal, please see our Privacy Policy

Reviewer #2: No

---

## [Editor Report · Decision Letter 2]

21 Dec 2025

Child's disability status and postnatal healthcare utilization among forcibly displaced women: Evidence from Pakistan

PGPH-D-25-02580R2

Dear Mr Dowou,

We are pleased to inform you that your manuscript 'Child's disability status and postnatal healthcare utilization among forcibly displaced women: Evidence from Pakistan' has been provisionally accepted for publication in PLOS Global Public Health.

Best regards,

Julia Robinson

Executive Editor